# Multiple Sclerosis in Mongolia; the First Study Exploring Predictors of Disability and Depression in Mongolian MS Patients

**Myadagmaa Jaalkhorol** [1,2,*], **Oyunbileg Dulamsuren** [1], **Amarsaikhan Dashtseren** [2,3], **Enkh-Amgalan Byambajav** [4], **Nansalmaa Khaidav** [5], **Badrangui Bat-Orgil** [6], **Anar Bold** [7], **Enkhdulguun Amgalan** [3], **Anujin Chuluunbaatar** [5] and **Ikuo Tsunoda** [8,*]

1    Division for Student Development and Management, Mongolian National University of Medical Sciences, Ulaanbaatar 14210, Mongolia
2    Mongolian Naran Society for Osteoarthritis and Musculoskeletal Health, Ulaanbaatar 14210, Mongolia
3    Department of Preventive Medicine, School of Public Health, Mongolian National University of Medical Sciences, Ulaanbaatar 14210, Mongolia
4    Department of Finance, Business School, National University of Mongolia, Ulaanbaatar 14210, Mongolia
5    Department of Health Social Work and Social Sciences, School of Public Health, Mongolian National University of Medical Sciences, Ulaanbaatar 14210, Mongolia
6    Department of Natural Sciences, Goethe High School, Ulaanbaatar 14210, Mongolia
7    Department of Neurosurgery, The General Hospital for State Special Servants, Ulaanbaatar 14210, Mongolia
8    Department of Microbiology, Faculty of Medicine, Kindai University, Osakasayama 589-8511, Japan
*    Correspondence: myadagmaa@mnums.edu.mn (M.J.); itsunoda@med.kindai.ac.jp (I.T.); Tel.: +976-999-50077 (M.J.); +81-72-366-0221 (I.T.)

**Abstract:** Mongolia is located at 45° north latitude in the center of the Asian continent, and about 80% of the territory is at 1000 m above sea level. Epidemiologically, multiple sclerosis (MS) has not been investigated in Mongolia, although there have been a few MS case reports. We investigated the characteristics of MS in Mongolia for the first time, focusing on the association between MS-related parameters and depression levels. We initiated cross-sectional analyses, using data from 27 MS patients aged 20 to 60 years in Ulaanbaatar, Mongolia. The patients completed a questionnaire on their lifestyles and clinical information. We classified the MS patients on the basis of disability levels using the expanded disability status scale (EDSS) scores: 11.1% mild disability and 88.9% moderate to severe disability (median EDSS score, 5.5). We also classified the patients on the basis of depression levels using the 9-item patient health questionnaire (PHQ-9) scores: 44.4% mild depression, 40.7% moderate depression, and 14.8% severe depression (mean PHQ-9's score, 9.96 ± 5.05). We used multivariate logistical regression analyses to identify predictors of EDSS or PHQ-9 scores. Disability levels were associated with vision and balance problems. Depression levels were associated with corticosteroid treatment; no patients were treated with disease-modifying drugs (DMDs). The odds ratios for disease onset age and treatment duration were associated with EDSS scores. In conclusion, MS onset age and treatment duration were independent predicting factors influencing the level of disability. Appropriate DMD treatment would lower the disability and depression levels.

**Keywords:** depression; disability; epidemiology; Mongolia; multiple sclerosis

## 1. Introduction

Multiple sclerosis (MS) is a chronic inflammatory demyelinating disease of the central nervous system (CNS) affecting about 400,000 people in the US [1] and 2.5 million people in the world [2]. MS is the major cause of non-traumatic neurologic disability in young adults [3,4]. Compared with other chronic diseases, MS ranks second only behind congestive heart failure in direct all-cause medical costs, with an estimated direct and indirect cost of $8528 to $54,244 per patient with MS annually [5].

MS is relatively uncommon in the Asia Pacific region. According to the "atlas of MS", the estimated prevalence of MS cases in China, Korea, Taiwan, South East Asia, and the Pacific region was between 0 and 5 per 100,000, and that in South Asia and Japan it was between 5 and 20 per 100,000 [6]. Although the above Asian countries have performed their own MS research for decades, MS has not been investigated in Mongolian people composed of different racial groups, including of Mongolian ancestry, except for only a few MS case reports [7]. There is no epidemiological or clinical research on the prevalence and characteristics of MS in Mongolia, and unfortunately, there are no available disease-modifying drugs (DMDs) in the country and insufficient information on the disease in general.

Mongolia is located at 45° north latitude in the center of the Asian continent, and about 80 percent of the territory is at an altitude of 1000 m above sea level; the climate of most Mongolian areas is categorized as cool region [8]. We anticipated that MS in Mongolia may have unique characteristics since MS has been reported to be more common in regions distant from the equator where there is less ultraviolet (UV) light. This finding has been associated with growing evidence suggesting that vitamin D may play a role in MS; vitamin D has been shown to regulate immune responses and affect the clinical signs of human MS and its animal model, experimental autoimmune encephalomyelitis (EAE) [9,10].

The disability status of MS measured by the expanded disability status scale (EDSS) has been associated with various factors/parameters, including cognitive and neuropsychiatric signs/symptoms, particularly depression. Depression has been reported to develop in 20–40% of MS patients [11]. In the current study, we mainly focused on the influence of depression on MS patients because of (1) a high prevalence of depression in Mongolia [12] and (2) the high risk of suicide in MS patients [13]. Previously, EDSS scores had been related to anxiety and depression factors [14]; disability level, depression, and fatigue were reported to be significant and independent predictors of disability in MS patients [15]. However, a recent meta-analysis showed that, although the prevalence of depression in MS patients was 27.01%, the prevalence of depression for patients with EDSS < 3 was 26.69% and for EDSS > 3 was 22.96% [16]. Many MS studies have demonstrated that overall well-being was not a simple manifestation of impairment or disability; for example, the quality of life of MS patients has been shown to differ, depending on their education level and employment status [17–19].

The epidemiologic characteristics of MS would be different between Mongolia and other countries due not only to the latitude and altitude of their residence but also to their lifestyle, food consumption, and nomadic culture. In the present study, we investigated the health-related disability and depression levels among MS patients, for the first time, in Mongolia. We found that disability levels were associated with vision and balance problems. Depression levels were associated with corticosteroid treatment; 48% of patients were treated with corticosteroid, but none of the patients received DMDs. MS onset age and treatment duration were independent predicting factors influencing the level of disability.

## 2. Materials and Methods

### 2.1. Study Design, Setting, Participants, and Questionnaire

We conducted a retrospective cross-sectional study (medical records based on control materials and identified each recorded case report) at the Health Units of Ulaanbaatar, Mongolia. We recruited a total of 27 MS patients (24 females and three males); 25 relapsing-remitting (RR)-MS, one secondary progressive (SP)-MS, and two primary progressive (PP)-MS from Ulaanbaatar's seven Health Units (Khan-Uul, Songinokhairkhan, Sukhbaatar, Bayangol, Bayanzurkh, Chingeltei, and Nalaikh). McDonald criteria [20] were utilized for the diagnosis of MS patients who belonged to the district hospital Ulaanbaatar and were over 20 years old. We excluded patients who had refused to participate in the survey, were under 20 years old, or did not belong to the Ulaanbaatar district.

Trained nurses conducted detailed interviews using a structured questionnaire. All the participants completed the questionnaire on their lifestyles (education, marital status,

employment, accommodation, smoking, alcohol, religion, and ethnicity) and clinical information (disease onset and duration, family history of MS, treatment, comorbidity, neurological signs including deep tendon reflex, vision, pyramidal and brainstem sign/symptom, sensation, disequilibrium, and urination) within 20–30 min from January to April 2022. All the participants provided written informed consent before participating in the study. The study protocol was approved by the Ethics Committee of the Mongolian National University of Medical Sciences (MNUMS, No.: 2021/0/12–2022/D-04).

### 2.2. EDSS and the 9-Item Patient Health Questionnaire (PHQ-9)

We conducted several neurological examinations, for example, evaluating reflex changes (patella tendon, achilleas tendon, and other deep tendon reflexes) and pyramidal signs (Babinski sign [21], spasticity, and spastic gait). Using the EDSS score (scores ranging from 0/normal to 10/death, in increments of 0.5) [22], neurologists quantified the disability levels of MS patients in eight functional systems (pyramidal, cerebellar, brainstem, sensory, bowel and bladder, visual, cerebral, and others) and assigned a functional system score in each of them. We classified the MS patients into two groups on the basis of the disability levels assessed by the EDSS: "mild disability," score < 5; and "moderate to severe disability," score ≥ 5 [23].

We assessed depression levels using the 9-item patient health questionnaire (PHQ-9); PHQ-9 is a 9-item self-report questionnaire in which participants were asked to rate how they felt in the previous 2 weeks. Each question was scored 0 to 3 (0, not at all; 1, several days; 2, more than half the days; and 3, nearly every day) with a resulting range of 0 to 27. We categorized depression levels using the PHQ-9 score: mild, 5–9; moderate, 10–14; and severe ≥15 [24,25].

### 2.3. Statistical Analyses

We used the Student's *t*-test for continuous variables and the chi-squared test for categorical variables to determine statistical differences in the subjects' gender, EDSS, and PHQ-9 scores. We used multivariate logistical regression analyses to identify predictors of EDSS or PHQ-9 scores. $p < 0.05$ was considered statistically significant. All the statistical analyses were performed with SPSS version 25.0 (SPSS Inc., Chicago, IL, USA).

## 3. Results

We recruited 27 MS patients whose mean age was 47.4 ± 1.6 years [mean ± standard deviation (SD)]. We classified the MS patients into two groups on the basis of the disability level assessed by the EDSS (median EDSS score, 5.5): 11.1% (three participants) with mild disability (score < 5) and 88.9% (24 participants) with moderate to severe disability (score ≥ 5). We also classified the patients into three groups on the basis of depression level using the PHQ-9 (mean PHQ-9's score, 9.96 ± 5.05): 44.4% (12 participants) with mild depression, 40.7% (11 participants) with moderate depression, and 14.8% (four participants) with severe depression.

First, we summarized sociodemographic characteristics (Table 1): education level, marital status, employment status, accommodation, smoking, alcohol consumption, religion, and ethnicity. We found that none of these was significantly associated with EDSS or PHQ-9 scores (data not shown).

**Table 1.** Sociodemographic characteristics of participants with multiple sclerosis (MS).

| Variable | Number | Percentage (%) |
|---|---|---|
| **Education Level** | | |
| high school | 9 | 33.3 |
| university | 18 | 66.7 |
| **Marital Status** | | |
| married | 22 | 81.5 |
| single | 4 | 14.8 |
| divorced/widowed | 1 | 3.7 |
| **Employment Status** | | |
| government organization | 1 | 3.7 |
| own business | 7 | 25.9 |
| retired/disability | 19 | 70.4 |
| **Accomodation** | | |
| apartment | 15 | 55.6 |
| house | 8 | 29.6 |
| ger * | 4 | 14.8 |
| **Lifestyle** | | |
| smoking | | |
| Yes | 5 | 18.5 |
| No | 22 | 81.5 |
| alcohol consumption | | |
| Yes | 2 | 7.4 |
| No | 25 | 92.6 |
| **Religion** | | |
| Buddhism | 12 | 44.4 |
| Shamanism | 6 | 22.2 |
| none | 9 | 33.3 |
| **Ethnicity** | | |
| Khalkh | 24 | 88.9 |
| Buryat | 2 | 7.4 |
| other | 1 | 3.7 |

*, traditional Mongolian dwelling.

Next, we examined whether the disability levels of MS patients determined by EDSS were associated with MS-related factors/parameters (Table 2). We found that the EDSS scores were significantly associated with older age ($p < 0.01$), vision problems ($p < 0.05$), and disequilibrium ($p < 0.01$), but not with the other factors/parameters. Although none of the patients were treated with DMDs, 13 patients were treated with corticosteroid, five patients were treated with methotrexate, and one patient was treated with plasmapheresis.

**Table 2.** Disability levels and MS-related factors/parameters.

| Variable | Number | Disability Level * | | *p* Value |
|---|---|---|---|---|
| | | Mild | Moderate/Severe | |
| **Age group** | | | | |
| 20–29 | 1 | 0 (0) | 1 (4.2) | |
| 30–39 | 3 | 1 (33.3) | 2 (8.3) | |
| 40–49 | 9 | 0 (0) | 9 (37.5) | |
| >50 | 14 | 2 (66.7) | 12 (50) | <0.01 |
| **MS onset age** | | | | |
| 20–29 | 8 | 1 (33.3) | 7 (29.2) | 0.962 |
| 30–39 | 11 | 1 (33.3) | 10 (41.7) | |
| 40–49 | 8 | 1 (33.3) | 7 (29.2) | |

**Table 2.** *Cont.*

| Variable | | Number | Disability Level * | | *p* Value |
|---|---|---|---|---|---|
| | | | **Mild** | **Moderate/Severe** | |
| **Family member with MS** | Yes | 4 | 0 (0) | 14 (14.8) | 0.444 |
| | No | 23 | 3 (100) | 20 (85.2) | |
| **MS disease duration** | 0–5 years | 6 | 1 (33.3) | 5 (20.8) | 0.885 |
| | 6–10 years | 11 | 1 (33.3) | 10 (41.7) | |
| | >11 years | 10 | 1 (33.3) | 9 (37.5) | |
| **MS treatment duration** | 0–5 months | 9 | 1 (33.3) | 8 (33.3) | 0.869 |
| | 6–12 months | 6 | 1 (33.3) | 5 (20.8) | |
| | >1 years | 12 | 1 (33.3) | 11 (45.8) | |
| **Treatment** | | | | | |
| Corticosteroid (oral) | Yes | 13 | 2 (67.7) | 11 (45.8) | 0.496 |
| | No | 14 | 1 (33.3) | 13 (54.2) | |
| Methotrexate (2.5 mg/day) | Yes | 5 | 2 (40) | 1 (20) | 0.381 |
| | No | 5 | 2 (40) | 1 (20) | |
| **History/comorbidity (%)** | | | | | |
| diabetes mellitus | Yes | 1 | 0 (0) | 1 (4.2) | 0.719 |
| | No | 26 | 3 (12) | 23 (95.8) | |
| cardiovascular disease | Yes | 5 | 0 (0) | 5 (20.8) | 0.381 |
| | No | 22 | 3 (14.2) | 19 (79.2) | |
| ophthalmic disease | Yes | 6 | 1 (33.3) | 5 (20.8) | 0.623 |
| | No | 21 | 2 (66.7) | 19 (79.2) | |
| rheumatic disease | Yes | 13 | 1 (33.3) | 12 (50) | 0.586 |
| | No | 14 | 2 (66.7) | 12 (50) | |
| **Neurological sign/symptom** | | | | | |
| abnormal deep tendon reflex | Yes | 5 | 0 (0) | 5 (20.8) | 0.381 |
| | No | 22 | 3 (100) | 19 (79.2) | |
| vision problem | Yes | 23 | 2 (67.7) | 21 (87.5) | <0.05 |
| | No | 4 | 1 (33.3) | 3 (12.5) | |
| pyramidal sign | Yes | 18 | 2 (11.1) | 16 (88.9) | 0.721 |
| | No | 9 | 1 (88.9) | 8 (11.1) | |
| Brainstorm sign/symptom | Yes | 3 | 0 (0) | 3 (12.5) | 0.516 |
| | No | 24 | 3 (100) | 21 (87.5) | |
| sensation problem | Yes | 17 | 2 (67.7) | 15 (62.5) | 0.888 |
| | No | 10 | 1 (33.3) | 9 (37.5) | |
| disequilibrium | Yes | 23 | 1 (4.3) | 22 (95.7) | <0.01 |
| | No | 4 | 2 (95.7) | 2 (4.3) | |
| urination problem | Yes | 14 | 1 (67.7) | 13 (54.2) | 0.681 |
| | No | 12 | 1 (33.3) | 11 (45.8) | |
| **Depression level (PHQ-9 score)** | mild (5–9) | 12 | 2 (66.3) | 10 (41.7) | 0.282 |
| | moderate (10–14) | 11 | 0 (0) | 11 (45.8) | |
| | severe (>15) | 4 | 1 (33.3) | 3 (12.5) | |

* Disability levels by the expanded disability status scale (EDSS): mild disability, EDSS score < 5; and moderate/severe disability, EDSS. score ≥ 5. Oral corticosteroid treatment was used as a long-term therapy (more than 5 months). Methotrexate (2.5 mg/day) was used long-term (more than 3 months). Abbreviations: MS, multiple sclerosis; PHQ, 9-item patient health questionnaire.

Then we examined whether the depression levels of the MS patients determined by PHQ-9 were associated with MS-related factors/parameters (Table 3). We found that depression levels were significantly associated with corticosteroid treatment ($p < 0.05$) but not with the other factors/parameters, including the EDSS scores.

**Table 3.** Depression levels and MS-related factors/parameters.

| Variables | | Number | Depression Level *: Number (%) | | | *p* Value ** |
|---|---|---|---|---|---|---|
| | | | Mild | Moderate | Severe | |
| **Age group** | 20–29 | 1 | 0 (0) | 1 (9.1) | 0 (0) | 0.132 |
| | 30–39 | 3 | 0 (0) | 2 (18.2) | 1 (25) | |
| | 40–49 | 9 | 3 (25) | 3 (27.3) | 3 (75) | |
| | >50 | 14 | 9 (75) | 5 (2) | 0 (0) | |
| **MS onset age** | 20–29 | 8 | 5 (41.7) | 1 (9.1) | 2 (50) | 0.942 |
| | 30–39 | 11 | 4 (33.3) | 5 (45.5) | 2 (50) | |
| | 40–49 | 8 | 3 (25) | 5 (45.5) | 0 (0) | |
| **Family member with MS** | Yes | 4 | 1 (8.3) | 3 (27.3) | 0 (0) | 0.294 |
| | No | 23 | 11 (91.7) | 8 (72.7) | 4 (100) | |
| **MS disease duration** | 0–5 years | 6 | 1 (8.3) | 4 (36.4) | 1 (25) | 0.156 |
| | 6–10 years | 11 | 4 (33.3) | 4 (36.4) | 3 (75) | |
| | >11 years | 10 | 7 (58.3) | 3 (27.3) | 0 (0) | |
| **MS treatment duration** | 0–5 months | 9 | 2 (16.7) | 6 (54.5) | 1 (25) | 0.408 |
| | 6–12 months | 6 | 3 (25) | 2 (18.2) | 1 (25) | |
| | >1 years | 12 | 7 (58.3) | 3 (27.3) | 2 (50) | |
| **Treatment** | | | | | | |
| Corticosteroid (oral) | Yes | 13 | 6 (50) | 3 (27.3) | 4 (100) | <0.05 |
| | No | 14 | 6 (50) | 8 (72.7) | 0 (0) | |
| Methotrexate (2.5 mg/day) | Yes | 5 | 2 (16.7) | 2 (18.2) | 1 (25) | 0.933 |
| | No | 22 | 10 (83.3) | 9 (81.8) | 3 (75) | |
| **History/comorbidity (%)** | | | | | | |
| diabetes mellitus | Yes | 1 | 0 (0) | 0 (0) | 1 (25) | 0.051 |
| | No | 26 | 12 (100) | 11 (100) | 3 (75) | |
| cardiovascular disease | Yes | 5 | 4 (33.3) | 0 (0) | 1 (25) | 0.113 |
| | No | 21 | 8 (66.7) | 11 (100) | 3 (75) | |
| ophthalmic disease | Yes | 6 | 2 (16.7) | 4 (36.6) | 0 (0) | 0.268 |
| | No | 21 | 10 (83.3) | 7 (63.4) | 4 (100) | |
| rheumatic disease | Yes | 13 | 7 (58.3) | 6 (54.6) | 0 (0) | 0.111 |
| | No | 14 | 5 (41.7) | 5 (45.4) | 4 (100) | |
| **Neurological sign/symptom** | | | | | | |
| abnormal deep tendon reflex | Yes | 5 | 3 (25) | 1 (9.1) | 1 (25) | 0.579 |
| | No | 22 | 9 (75) | 10 (90.9) | 3 (75) | |
| vision problem | Yes | 23 | 11 (91.7) | 9 (81.8) | 3 (75) | 0.661 |
| | No | 4 | 1 (8.3) | 2 (18.2) | 1 (25) | |
| pyramidal sign | Yes | 12 | 9 (50) | 6 (54.5) | 3 (75) | 0.541 |
| | No | 9 | 3 (33.3) | 5 (55.5) | 1 (25) | |
| brainstem sign/symptom | Yes | 3 | 1 (8.3) | 1 (9.1) | 1 (25) | 0.631 |
| | No | 24 | 11 (91.7) | 10 (90.9) | 3 (75) | |
| sensation problem | Yes | 17 | 7 (58.3) | 6 (54.5) | 4 (100) | 0.297 |
| | No | 10 | 5 (41.7) | 5 (45.5) | 0 (0) | |
| disequilibrium | Yes | 23 | 10 (83.3) | 10 (90.9) | 3 (75) | 0.724 |
| | No | 4 | 2 (16.7) | 1 (9.1) | 1 (25) | |
| Urination problem | Yes | 15 | 7 (58.3) | 4 (36.4) | 4 (100) | 0.087 |
| | No | 12 | 5 (41.7) | 7 (63.6) | 0 (0) | |
| **EDSS score** | <5 | 3 | 2 (16.7) | 0 (0) | 1 (25) | 0.282 |
| | ≥5 | 24 | 10 (83.3) | 11 (100) | 3 (75) | |

* We categorized depression levels using the PHQ-9 score: mild, 5–9; moderate, 10–14; and severe ≥15. ** *p* values were obtained by chi-squared test. Oral corticosteroid treatment was used as a long-term therapy (more than 5 months). Methotrexate (2.5 mg/day) was used long-term (more than 3 months). Abbreviations: MS, multiple sclerosis; EDSS, expanded disability status scale.

Lastly, we determined age-adjusted odds ratios (ORs) for EDSS scores and other covariates; age, gender, MS onset age, MS treatment duration, and depression severity. Table 4 shows significant unadjusted ORs for EDSS scores. After adjusting for age, OR for disease onset age remained significantly associated with EDSS scores [OR, 1.50; *p* < 0.001;

95% confidence interval (CI) (1.02–4.39)]; OR for treatment duration also had a significant effect on the EDSS scores [OR, 1.02; *p* < 0.01; 95% CI (1.00–1.17)].

**Table 4.** Unadjusted ORs and age-adjusted ORs for EDSS scores and other covariates.

| Variable | Unit/Category | Unadjusted | | | Age-Adjusted | | |
|---|---|---|---|---|---|---|---|
| | | OR | 95% CI | *p*-Value | OR | 95% CI | *p*-Value |
| age | year | 1.00 | 1.00–1.34 | <0.001 | - | - | - |
| gender | female/male | 1.03 | 1.02–1.18 | 0.707 | 1.00 | 1.00–1.17 | 0.743 |
| MS onset age | year | 1.50 | 1.21–4.75 | <0.001 | 1.50 | 1.02–4.39 | <0.001 |
| MS treatment duration | year | 1.18 | 1.30–4.60 | <0.05 | 1.02 | 1.00–1.17 | <0.01 |
| depression severity | score | 1.00 | 0.78–1.27 | 0.88 | 1.30 | 0.19–4.64 | 0.786 |

Abbreviations: CI, confidence interval; EDSS, Expanded Disability Status Scale; OR, odds ratio.

## 4. Discussion

In this study, for the first time, we reported the characteristics of MS patients in Mongolia, in particular exploring predictors of disability and depression in MS. Several environmental factors, including geographical latitude, have been attributed to the etiology of MS [26]. The participants in this study were citizens of Ulaanbaatar, the capital and most populous city of Mongolia; since 1991, migration of Mongolians from rural to urban areas has increased [27]. Ulaanbaatar is located in a valley of the Bogd mountain on the Tuul river with an elevation of 1350 m. Ulaanbaatar has an average annual temperature of −0.4 °C, making it the coldest capital in the world; heavy snowfall and cold temperatures last for 3 months [28]. In winter, Ulaanbaatar is one of the cities with the worst air pollution in the world, which is caused by the burning of raw coal and waste by people living in low-income neighborhoods. Although Ulaanbaatar banned the use of raw coal in 2019, air pollution has not been reduced. Previously, urbanization and air pollution levels have been related to MS [29,30].

In this study, we found no association between the sociodemographic characteristics of the MS patients and disability/depression levels. For example, in Table 1, we found that our participants' smoking prevalence was 18.5% (5 of 27 participants); smoking was not significantly associated with EDSS or PHQ-9 scores (data not shown), although smoking has been associated with MS prognosis [31,32].

Among MS-related factors and parameters, we found that vision and balance problems were significantly associated with disability levels (Table 2). Interestingly, the MS prevalence in family members of our MS patients was 14.8% (4 of 27 participants). Although we found no association between familial MS and EDSS in Table 2, it has been reported that the EDSS score was higher in the group of familial cases and that a family history of MS was correlated with EDSS scores [33–35]. In Table 2, we showed that our participants' rheumatic disease prevalence was high (48%, 13 of 27 participants); according to the Mongolian Health Indicators of the population, the prevalence of rheumatic disease in Mongolia was 4.0 per 10,000 in 2019. We also found that the prevalence of rheumatic disease was not significantly associated with EDSS scores or depression levels in our study, although rheumatoid arthritis comorbidity has been reported to be statistically significant in MS patients [36].

Depression is one of the most common comorbidities in MS [37]. Depression in MS can be due to disabilities or treatment including steroids and interferon-β. In Table 3, we found that corticosteroid treatment was significantly associated with depression levels by the PHQ-9. This is consistent with previous reports showing the relationship between depression and corticosteroid treatment in MS patients [38,39]; depression has been reported as one of the side effects of corticosteroid treatment [40,41]. In Mongolia, DMDs, such as natalizumab and fingolimod, are not available, although DMDs have been shown to reduce clinical and MRI disease activity compared with corticosteroid and other drugs used in Mongolia [42–47]. Treatment with appropriate DMDs would lower not only the disability

levels but also the incidence of depression in MS patients in Mongolia, avoiding the usage of corticosteroid.

In Table 3, we found no association between age and depression levels in MS patients. This is consistent with a previous study showing that depression level in MS was not significantly associated with the older age group [48]. Intriguingly, in our study, only one patient had diabetes mellitus, although a Danish study reported a threefold-high incidence rate for MS in the population with type 1 diabetes [49,50]. Furthermore, diabetes mellitus was statistically higher among older-age patients with MS (OR = 1.31) [51]; diabetes mellitus and anxiety have been associated with cognitive dysfunction in MS [52].

In Table 4, we demonstrated that MS onset age was a predicting factor influencing the level of disability; the MS onset age of all the participants in our study was after 20 years of age. Our result was consistent with previous MS studies examining the association between onset age and disability levels. For example, MS onset age has been reported to be significantly associated with severe disability determined by the EDSS [48,53–56]. On the other hand, Ghezzi [57] reviewed and summarized the clinical data of MS with onset before 16 years of age (early-onset MS or juvenile MS), and concluded that early-onset MS reached mild and severe disability levels after a longer time but at lower age, compared with adult-onset MS cases. Similarly, Mirmosayyeb et al. [58] reported that, in early-onset MS, 14.2% with RR-MS were diagnosed with SP-MS on average 24.6 years after disease onset and that, in adult-onset MS, 15.6% with RR-MS progressed to SP-MS on average 20.5 years after the disease onset.

One of the factors that may play a role in the MS pathogenesis unique in Mongolia is UV radiation. Although Mongolia is located at an altitude of 1000 m above sea level, the UV radiation is high in June and July in Mongolia [8,59]. Mongolia is located at a high latitude (the city, Ulaanbaatar, is 48° N), where the amount of UV rays reaching the earth's surface is insufficient November to April [60]. People living at high latitudes are more prone to vitamin D deficiency, especially in winter, due to the oblique angle of the sun's zenith. It has also been proven that people living above and below 33° latitude have little or no vitamin D production in the skin during the winter [61]. Furthermore, in Mongolia, there is a lack of foods rich in vitamin D (e.g., foods, including fatty fish, are rich in natural vitamin D) [62]. Although sun exposure and vitamin D may have independent risk factors for CNS demyelination [63], a recent meta-analysis reported [64] a significant association between low vitamin D and increased risk of developing MS. Although a high prevalence of vitamin D deficiency has been reported in Mongolia [65], serum vitamin D levels in MS patients were not performed in our current study.

Our study has limitations worth noting. First, the participants were not selected randomly from different geographic areas in Mongolia; all the participants were Ulaanbaatar citizens. Thus, this survey did not represent the general Mongolian population. Second, dietary and lifestyle factors have been associated with MS patients in the cohort [66] and case-control studies [67]. Mongolia is a country with considerable ethnic and nomadic cultural diversity; for example, the Khalkh, Buryat, Torguud, Bayad, Zakhchin, Buriad, Kazakh, Dariganga, and Uriankhai people have large differences in lifestyles, nutrition, and environmental factors [68]. However, most of the participants in our study were Khalkh (24 of 27 participants), and we did not ask about a variety of lifestyle factors unique to each ethnicity in the survey of MS patients. Third, we did not obtain more detailed information about clinical and immunological data, such as neuroimaging and oligoclonal bands. Forth, although only corticosteroid treatment was significantly associated with depression levels in our study, other causations and cofounders should be assessed to identify the true causative factors. Future treatment with DMDs, instead of corticosteroids, would give an insight into the role of corticosteroid treatment in the depression of MS in Mongolia.

## 5. Conclusions

We demonstrated that MS onset age and treatment duration were independent predicting factors influencing the level of disability in Mongolia. In the current study, about half of the MS patients were treated with corticosteroid, which was associated with depression levels; none of the patients were treated with DMDs. The availability of appropriate DMDs in Mongolia would lower the disability and depression levels of MS patients.

**Author Contributions:** Conceptualization, M.J. and I.T.; methodology, M.J., I.T., O.D., A.D. and A.C.; software, B.B.-O. and E.-A.B.; validation, M.J., B.B.-O. and E.-A.B.; formal analysis, M.J. and B.B.-O.; investigation, M.J., A.B. and A.C.; resources, A.D.; data curation, M.J. and N.K.; writing—original draft preparation, M.J.; writing—review and editing, I.T.; visualization, O.D. and E.A.; supervision, M.J. and I.T.; project administration, M.J. and I.T.; funding acquisition, I.T. All authors have read and agreed to the published version of the manuscript.

**Funding:** M.J., O.D., A.D., E.-A.B., N.K., B.B.-O., A.B., E.A. and A.C. received no specific grant from any funding agency in the public, commercial, or not-for-profit sectors. I.T. was supported by grants from the National Institute of General Medical Sciences COBRE Grant (P30-GM110703) and the KAKENHI from the Japan Society for the Promotion of Science [Grant-in-Aid for Scientific Research (C): JP20K07455].

**Institutional Review Board Statement:** The study protocol was approved by the Ethics Committee of the Mongolian National University of Medical Sciences (MNUMS, No.: 2021/3/12–2022/D-04).

**Informed Consent Statement:** Informed consent was obtained from all the subjects involved in the study.

**Data Availability Statement:** The data presented in this study are available on request from the corresponding author. The data are not publicly available due to privacy and ethical reasons.

**Acknowledgments:** The authors thank the personnel from the Department of Neurology, Chingeltei Health Unit, Songinokhairkhan Health Unit, Bayanzurkh Health Unit, Khan-Uul Health Unit, Baganuur Health Unit, Division for Student Development and Management, Mongolian National University of Medical Sciences, Ulaanbaatar, Mongolia, and Department of Microbiology, Kindai University Faculty of Medicine, Osaka, Japan, for their technical assistance with the surveys. We also thank all the study participants.

**Conflicts of Interest:** The authors declare no conflict of interest.

## Abbreviations

| | |
|---|---|
| CNS | Central nervous system |
| DMD | Disease-modifying drug |
| EAE | Experimental autoimmune encephalomyelitis |
| EDSS | Expanded disability status scale |
| MNUMS | Mongolian National University of Medical Sciences |
| MS | Multiple sclerosis |
| PHQ-9 | 9-item patient health questionnaire |
| PP | Primary progressive |
| RR | Relapsing-remitting |
| SP | Secondary progressive |
| UV | Ultraviolet |

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
