# Peer review of "Multiple Sclerosis in Mongolia; the First Study Exploring Predictors of Disability and Depression in Mongolian MS Patients"

_pathophysiology, doi:10.3390/pathophysiology30010003_

Round 1

Reviewer 1 Report

The fact this is the first report addressing certain aspects of multiple sclerosis (MS) in a region of the world with a notable very low prevalence, deserves to be commended. The paper however, leaves the reader with a number of unanswered questions and doubts which require further clarification.

(1) Line 53, the medical costs are given in USD and that should be emphasized. (2) There is no epidemiological or clinical research on MS in Mongolia, and unfortunately no available DMDs in the country, and perhaps insufficient information on the disease in general. Nevertheless, (3) the manuscript does not mention the criteria utilized for MS diagnosis. Twenty one patients (out of 27) had a duration of disease longer than 6 years hence, the 2017 McDonald version did not apply for the majority. (4) How corticosteroid therapy was used in almost half of the patients: orally? chronically? (5) The methotrexate dose is not mentioned either. (6) Only one patient received plasmapheresis. I would suggest not to include it in table 2 but would mentioned it in the text. (7) I would not include in the table amitryptiline and diazepam since these are symptomatic therapies, not MS therapies. (8) Your discussion on vitamin D deficiency is noted, however, this theoretical assumption is not proven in your cases since none had a serological vitamin D level performed, or at least not mentioned in your paper. 

Author Response

Responses to Comments by Reviewer #1

General Comment

The fact this is the first report addressing certain aspects of multiple sclerosis (MS) in a region of the world with a notable very low prevalence, deserves to be commended. The paper however, leaves the reader with a number of unanswered questions and doubts which require further clarification.

Response to general comment

We are grateful to the reviewer for evaluating our study as intriguing, and also greatly appreciate highly relevant and helpful comments and suggestions from the reviewer. We have revised the manuscript, addressing all comments raised by the reviewer in point-by-point discussion as follows:

Comments

  1. Line 53, the medical costs are given in USD and that should be emphasized.

We appreciate the comment. Accordingly, we added the following sentences:

On page 2, lines 54-56

“Compared with other chronic diseases, MS ranks second only behind congestive heart failure in direct all-cause medical costs, with an estimated direct and indirect cost of $8,528 to $54,244 per patient with MS annually”

  1. There is no epidemiological or clinical research on MS in Mongolia, and unfortunately no available DMDs in the country, and perhaps insufficient information on the disease in general. Nevertheless,

We agree with the comment.  Accordingly, we have rewritten the sentences in the Introduction as follows:

Old

There is no epidemiological or clinical research on the prevalence and characteristics of MS in Mongolia.

New

On page2, lines 62-63

There is no epidemiological or clinical research on the prevalence and characteristics of MS in Mongolia, and unfortunately no available disease modifying drugs (DMDs) in the country, as well as insufficient information on the disease in general.

  1. The manuscript does not mention the criteria utilized for MS diagnosis. Twenty one patients (out of 27) had a duration of disease longer than 6 years hence, the 2017 McDonald version did not apply or the majority.

To address the concern, we added the following sentences with a new reference:

On page 2, line 95 (Method section)

McDonald’s criteria [20] was utilized for diagnosis of MS patients.

New reference:

Thompson, A. J.; Banwell, B. L.; Barkhof, F.; Carroll, W. M.; Coetzee, T.; Comi, G.; Correale, J.; Fazekas, F.; Filippi, M.; Freedman, M. S., Diagnosis of multiple sclerosis: 2017 revisions of the McDonald criteria. The Lancet Neurology 2018, 17 (2), 162-173.

  1. How corticosteroid therapy was used in almost half of the patients: orally? chronically?

We have added “oral” in Tables 2 and 3 as follows:

Corticosteroid (oral)  

  1. The methotrexate dose is not mentioned either.

We have added the dose “2.5 mg/day” in Tables 2 and 3.

  1. Only one patient received plasmapheresis. I would suggest not to include it in table 2,3 but would mentioned it in the text.

We agree with the comment. Accordingly, in the revision, we included plasmapheresis treatment information in the text, but not in Tables and 3 as follows:

Old

About half of the patients were treated with corticosteroid and methotrexate; no patients were treated with DMDs.

New

On page 4, lines 149-151

Although no patients were treated with DMDs, 13 patients were treated with corticosteroid, five patients were treated with methotrexate, and one patient was treated with plasmapheresis.

  1. I would not include in the table amitriptyline and diazepam since these are symptomatic therapies, not MS therapies.

We agree with the comment and have excluded “amitryptiline and diazepam” from Tables 2 and 3.

  1. Your discussion on vitamin D deficiency is noted, however, this theoretical assumption is not proven in your cases since none had a serological vitamin D level performed, or at least not mentioned in your paper. 

We appreciate the comment. Although a high prevalence of vitamin D deficiency has been reported in Mongolia, we have not examined a serological vitamin D level in the current study.  We have included the following sentences with a new reference as follows:

On page 9, lines 233-235 (Discussion section)

Although a high prevalence of vitamin D deficiency has been reported in Mongolia (Bater et al., 2021), serum vitamin D levels in MS patients were not performed in our current study.

New reference

Bater J, Bromage S, Jambal T, Tsendjav E, Lkhagvasuren E, Jutmann Y, Martineau AR, Ganmaa.D. Prevalence and Determinants of Vitamin D Deficiency in 9595 Mongolian Schoolchildren: A Cross-Sectional Study. Nutrients. 2021 Nov 21;13(11):4175. doi: 10.3390/nu13114175.

Reviewer 2 Report

I reviewed the article entitled "Multiple sclerosis in Mongolia; the first study exploring predictors of disability and depression in Mongolian MS patients". 

There are some remarks which should be considered: 

1- The abstract does not have any conclusion. 

2- More details should be reported in the abstract, including the EDSS median (IQR) and PHQ-9's mean (SD). Moreover, the inclusion and exclusion criteria could be reported.

3- The keywords don't match the title and aims of the studies. 

4- What was the diagnostic criteria that MS patients were diagnosed with?

5- The analysis and assessments that have been taken are not sufficient. MS patients suffer from many problems, including but not limited to depression. What was the reason that only depression is assessed?

6- It is not a true decision that the level of depression links to corticosteroid use. Depression is a common presentation among MS patients, and to assess the causations, more cofounders and factors should be considered to make a correct conclusion. 

Author Response

Responses to Comments by Reviewer #2

General Comment

I reviewed the article entitled "Multiple sclerosis in Mongolia; the first study exploring predictors of disability and depression in Mongolian MS patients". 

There are some remarks which should be considered: 

Response to general comment

We are grateful to the reviewer for reviewing our manuscript.  We have addressed all concerns raised by the reviewer as follows:

Comments

  1. The abstract does not have any conclusion. 

Accordingly, we added the following sentences in the abstract:

On page 1 lines 46-48 (Abstract)

In conclusion, MS onset age and treatment duration were independent predicting factors influencing the level of disability. Appropriate DMD treatment would lower the disability and depression levels.

  1. More details should be reported in the abstract, including the EDSS median (IQR) and PHQ-9's mean (SD). Moreover, the inclusion and exclusion criteria could be reported.

To address the comment, we have included information about median EDSS and mean PHQ-9’s score in the abstract, and the inclusion and exclusion criteria in the Method section with a new reference as follows:

On page 1, line 39 and line 41

(median EDSS score, 5.5) (mean PHQ-9’score, 9.96 ± 5.05)

On page 2, lines 95-97 (Method section)

McDonald criteria [20] was utilized for diagnosis of MS patients, who belonged to the district hospital Ulaanbaatar, and were over 20 years old. We excluded the patients, who had refused to participate in the survey, under 20 years old, or citizens who did not belong to the district Ulaanbaatar.

New reference:

Thompson, A. J.; Banwell, B. L.; Barkhof, F.; Carroll, W. M.; Coetzee, T.; Comi, G.; Correale, J.; Fazekas, F.; Filippi, M.; Freedman, M. S., Diagnosis of multiple sclerosis: 2017 revisions of the McDonald criteria. The Lancet Neurology 2018, 17 (2), 162-173.

  1. The keywords don't match the title and aims of the studies.

Accordingly, we have changed the keywords as follows:

On page 1, line 40

Keywords: depression, disability, epidemiology, Mongolia, multiple sclerosis

  1. What was the diagnostic criteria that MS patients were diagnosed with?

On page 2, line 95 (Method section)

McDonald criteria [20] was utilized for diagnosis of MS patients

New reference:

Thompson, A. J.; Banwell, B. L.; Barkhof, F.; Carroll, W. M.; Coetzee, T.; Comi, G.; Correale, J.; Fazekas, F.; Filippi, M.; Freedman, M. S., Diagnosis of multiple sclerosis: 2017 revisions of the McDonald criteria. The Lancet Neurology 2018, 17 (2), 162-173.

  1. The analysis and assessments that have been taken are not sufficient. MS patients suffer from many problems, including but not limited to depression. What was the reason that only depression is assessed?

We appreciate the comment.  Among cognitive and neuropsychiatric issues in MS, depression is one of the most serious problems in MS patients and is known as a risk factor for suicide.  In Mongolia, the prevalence of depression is known to be high; thus, we mainly focused on depression in the current study.  We have added the following sentences with new references:

On page 2, lines 72-75 (Introduction section)

The disability status of MS measured by the expanded disability status scale (EDSS) has been associated with various factors/parameters, including cognitive and neuropsychiatric signs/symptoms, particularly depression. Depression has been reported to develop in 20-40% of MS patients  [11]. In the current study, we mainly focused on the influence of depression on MS patients because of 1) a high prevalence of depression in Mongolia [12] and 2) the high risk of suicide in MS patients [13]

New references:

Patten SB, Marrie RA, Carta MG. Depression in multiple sclerosis. Int Rev Psychiat 2017; 29: 463–472.

Tsogbadrakh, B.; Yanjmaa, E.; Badamdorj, O.; Choijiljav, D.; Gendenjamts, E.; Ayush, O.-e.; Pojin, O.; Davaakhuu, B.; Sukhbat, T.; Dovdon, B., Frontline Mongolian Healthcare Professionals and Adverse Mental Health Conditions During the Peak of COVID-19 Pandemic. Frontiers in Psychology 2022, 1006.

Turner AP, Alschuler KN, Hughes AJ, et al. Mental health comorbidity in MS: Depression, anxiety, and bipolar disorder. Curr Neurol Neurosci Rep 2016; 16(12): 106. 

  1. It is not a true decision that the level of depression links to corticosteroid use. Depression is a common presentation among MS patients, and to assess the causations, more cofounders and factors should be considered to make a correct conclusion.

We agree with the comment; this is another limitation of our study.  We have included the following sentences:

Page 9, lines 244-247

Forth, although only the corticosteroid treatment was significantly associated with the depression levels in our study, other causations and cofounders should be assessed to identify the true causative factors.  Future treatment with DMDs, instead of corticosteroids, would give an insight into the role of corticosteroid treatment in the depression of MS in Mongolia.    

Round 2

Reviewer 1 Report

Thank you for addressing the questions and suggestions posed by this reviewer. The paper is now improved in content and structure. There is one item that in my opinion needs further clarification since it does have: you mention the corticosteroid treatment was "oral" and also, that development of depression levels were associated to this therapy. You also provide the dose of oral methotrexate 2.5 mg. Readers with a clinical interest would have reasonable questions to further clarify these items:

1. I assume oral corticosteroid treatment in your series was not used to treat acute attacks, but rather as a long-term therapy in the absence of DMT availability. If this is the case, a range of time utilization (weeks?, months?, years?) it would be helpful to have information on this aspect. If oral steroids were just utilized for management of relapses, this should be stated. Chronic use of steroids is known to be associated to depression and other health problems.

2. Oral methotrexate has also been used long-term in MS in regions of the world where access to DMT is scarce or unavailable. The question is if this medication was used in your series long-term as well.   

Author Response

Responses to Comments by Reviewer #1

General Comment

Thank you for addressing the questions and suggestions posed by this reviewer. The paper is now improved in content and structure. There is one item that in my opinion needs further clarification since it does have: you mention the corticosteroid treatment was "oral" and also, that development of depression levels were associated to this therapy. You also provide the dose of oral methotrexate 2.5 mg. Readers with a clinical interest would have reasonable questions to further clarify these items:

Response to general comment

We appreciate highly relevant and helpful comments from the reviewer. We have revised the manuscript, addressing all two comments raised by the reviewer in point-by-point discussion as follows:

Comments

  1. I assume oral corticosteroid treatment in your series was not used to treat acute attacks, but rather as a long-term therapy in the absence of DMT availability. If this is the case, a range of time utilization (weeks?, months?, years?) it would be helpful to have information on this aspect. If oral steroids were just utilized for management of relapses, this should be stated. Chronic use of steroids is known to be associated to depression and other health problems.

As this reviewer assumed, oral corticosteroid treatment was neither to treat acute attacks nor relapses, but was used as a long-term therapy (more than 5 months).  We have added this information to Tables 2 and 3 as follows:

Oral corticosteroid treatment was used as a long-term therapy (more than 5 months).

  1. Oral methotrexate has also been used long-term in MS in regions of the world where access to DMT is scarce or unavailable. The question is if this medication was used in your series long-term as well.   

Again, this reviewer’s assumption was correct.  Methotrexate has been used long-term in Mongolia. Accordingly, we have added this information to Tables 2 and 3 as follows:

Methotrexate (2.5 mg/day) was used long-term (more than 3 months).

Reviewer 2 Report

All of my comments are addressed. Thanks

Author Response

 "Thank your for the review"